# Child-Sensitive WASH Composite Score and the Nutritional Status in Cambodian Children

**DOI:** 10.3390/nu11092142

**Published:** 2019-09-07

**Authors:** Giulia Manzoni, Arnaud Laillou, Chea Samnang, Rathmony Hong, Frank T. Wieringa, Jacques Berger, Etienne Poirot, Francesco Checchi

**Affiliations:** 1Faculty of Epidemiology and Population Health, London School of Hygiene & Tropical Medicine, London WC1E 7HT, UK; 2United Nations Children’s Fund (UNICEF), Integrated Early Childhood Development, Exchange Square, 5th floor, No. 19&20, Street 106, Sangkat Wat Phnom, Khan Daun Penh, Phnom Penh P.O. Box 176, Cambodia; 3Council of Agriculture and Rural Development, Phnom Penh 12000, Cambodia; 4Institute of Research for Development (IRD), UMR Nutripass IRD/UM/SupAgro, 34394 Montpellier, France

**Keywords:** weight-for-height z-score, mid-upper arm circumference, height-for-age z-score, WASH, child-sensitive, WASH composite score

## Abstract

Progress in health has occurred in the past decades in Cambodia, in terms of health service access and interventions, but several indicators, including the prevalence of malnourished children, remain alarming. The causes of undernutrition are often linked to inadequate access to water, sanitation and hygiene services but limited evidence exists on the direct association between poor WASH practices and children’s’ nutritional statuses. This study investigates the relationship between water, sanitation and hygiene practices, defined as the child-sensitive composite score, and the nutritional status of children under five years old, measured as the weight-for-height z-score, mid-upper arm circumference or height-for-age z-score in six districts of Cambodia. The analysis used data from a longitudinal study, comprising extensive data collection on anthropometry, health, nutrition, WASH, and cognitive development. Chronological trends in wasting and stunting were described cross-sectionally, whereas the effect of WASH practices on the nutritional status of children over up to three consecutive study visits was examined with a linear mixed-effects model. The prevalence of wasting decreased during the study while stunting prevalence increased. A small, but significant, association was found between the WASH child-sensitive composite scores and the wasting child anthropometry indicators: weight-for-height z-score or mid-upper arm circumference. Evidence for an association with height-for-age z-score, detecting stunted children, was found when the independent variable was quantified according to global, but not national, guidelines. This study reinforces discordant existing evidence towards a direct association between WASH practices and children’s nutritional status, suggesting the need to align nutrition and WASH programmes.

## 1. Introduction

Despite progress over the past two decades, global child undernutrition rates remain alarmingly high, particularly in South Asia and in Sub-Saharan Africa. Stunting and wasting represent the predominant forms of undernutrition: stunting, a height for age two standard deviations below the mean, associated with chronic malnutrition, was globally estimated to affect 151 million children under five years old in 2017; wasting or acute malnutrition, defined as either a weight for height two standard deviations below the mean or a mid-upper arm circumference less than 125 mm, affected 51 million children in 2017 [1]. In Cambodia, where this study was carried out, the most recent Demographic Health Survey estimated the overall prevalence of undernutrition among children aged 0–59 months to be 32.4% for stunting and 9.8% for wasting. However, there were disparities between urban and rural provinces: 33.8% of rural children were stunted and 7.5% were wasted compared with 23.7% and 9.9% of urban children, respectively. By province, stunting prevalence was higher in Ratanakiri/Mondolkiri (39.8%) and Kratie (38.4%) than Phnom Penh (17.9%) while wasting prevalence was 8.4% in Phnom Penh, 8.2% in Ratanakiri/Mondolkiri and 6.5% in Kratie [2].

Undernutrition has a complex aetiology with numerous and overlapping factors, leading to short- and long-term consequences, such as child’s and mother’s morbidity and mortality, child’s physical and neurodevelopmental impairment and reduced economic performance [3,4,5,6,7]. Immediate causes include inadequate dietary intake and diseases; indirect factors comprise poverty, feeding and care practices, poor access to and low-quality health services and sociocultural, economic and political context [8] and are often liked to inadequate access to water, sanitation and hygiene (WASH) services [9]. However, most data shows associations between poor WASH practices and the incidence of diarrhoea or other infections rather than direct links to a child’s nutritional status. Diarrhoea, the main cause of mortality and morbidity among children under five years of age, has a bidirectional relationship with undernutrition: it can lead to undernutrition because it causes loss of appetite, lack of absorption and increased metabolism and, at the same time, undernutrition weakens the immune system of undernourished children, thereby increasing their susceptibility to infectious diseases [10,11]. Parasitic and helminthic infections have been shown to be directly linked to poor sanitation conditions [12], and can indirectly cause loses of nutrients, or impair absorption of nutrients. Moreover, helminth infection might affect child’s immunity and immune responses [13,14]. A systematic review and meta-analysis on the direct effects of WASH interventions on child’s growth identified five cluster-randomised controlled trials (RCTs), one occurred in Cambodia, reporting that solar disinfection of drinking water, provision of soap, and improvement of water quality slightly but significantly benefit child’s linear growth [13]. Furthermore three RCTs in Indonesia, India and Mali, showed that sanitation interventions reduced stunting or improved child height and weight [15,16,17] and a non-randomised study reported increased height-for-age z-scores (HAZ) in villages supported with WASH interventions including water quantity, sanitation and hygiene [18].

Nevertheless evidence in the literature includes discordant results. Two recent RCTs in Bangladesh and Kenya concluded that integration of water, sanitation, and handwashing with nutrition did not have any benefit on improved child’s growth [19,20]. One study in India showed that improved sanitation facilities decreased open defecation but did not improve children’s growth [21]. Furthermore three non-randomized trials targeting water quality and sanitation or hygiene promotions did not report any association with stunting [22,23,24].

Limited evidence exists in the literature for Cambodia where, in 2015, 30% of the rural population lacked basic drinking water, 51% used open defecation and 40% did not have basic hygiene services [25]. Child undernutrition has serious public health consequences for the Cambodian population and for the country’s economic development. Growth impairment in the first years of life leads to adverse health complications and to long-term economic hardships due to heavy health expenditures and decrease in productivity. In 2013–2014 the Royal Government of Cambodia, supported by UNICEF and the World Food Programme, estimated a loss of more than $400 million due to child malnutrition [26].

To understand the causes of this impoverishment UNICEF, in collaboration with the Institute of Research for Development (IRD) and Reproductive and Child Health Alliance (RACHA), performed a longitudinal study in one district in Phnom Penh, and five districts in Kratie and Ratanakiri, north-eastern provinces, to monitor the health and nutritional status of children and provide rapid feedback about the progress that can be made with enhanced health monitoring.

To date there is no evidence for a significant association between WASH and child nutrition status in South East Asian countries. In this paper we aim to investigate for the first time in Cambodia the longitudinal association between water, sanitation and hygiene practices, regrouped as child-sensitive WASH composite scores, and the nutritional status of children under five years of age measured as weight-for-age z-score (WHZ) and mid-upper arm circumference (MUAC) for wasting and height-for-age z-score (HAZ) for stunting.

## 2. Key Messages

Stunting prevalence remains alarmingly high in children living in targeted district of Cambodia.

A strong significant association was found between water, sanitation and hygiene conditions and wasting and stunting highlighting the needs to align the two sectors for reducing child’s undernutrition in Cambodia.

Further randomized controlled trials are needed to measure the impact of WASH interventions for each component on undernutrition in Cambodia.

## 3. Methods

### 3.1. Data Source

A secondary data analysis was performed using data collected in a longitudinal cohort study (“MyHealth” project) aiming to collect in-depth health and child’s nutritional status data for at least three years to better inform the government on progress that can be made with enhanced health monitoring. A total of six districts over three provinces were conveniently selected: Russei Kaev in Phnom Penh, AD Chitr Borie and Krong Kratie in Kratie, and AD Ou Chum, Krong Ban Lung and Bar Kaev in Ratanakiri province. At baseline a sample size of 1200 children under two years of age per province was calculated, to observe a reduction in child stunting from 32% to 26% over a 3–5-year period (with a precision of 3% and a dropout of 20%). A list of all the children under two years and pregnant women was obtained from midwives and village health volunteers for all the villages within the targeted districts. All pregnant women and mothers or caretakers of all children in the list were invited to participate to the longitudinal study. All participants provided written informed consent at baseline before their inclusion in the cohort study. At each visit, women were invited to central points, such as someone’s house, a pagoda, or other village gathering point, pre-arranged by the local village health volunteers. Enumerator teams, using electronic tablet-based questionnaire, collected information on (i) socio-economic status; (ii) health knowledge; (iii) diet, including dietary diversity of children and women and breastfeeding practices; (iv) WASH practices; (v) mother and child anthropometry and (vi) cognitive development. Baseline data were collected in April–May 2016 followed by six consequent follow-ups that will run until March 2019. This paper used data for children (*n* = 5328) collected at baseline, follow-up 2 occurred in October–November 2016 and follow-up 3 occurred in April–May 2017, including WASH practice information. The longitudinal study was ethically approved on February 2016 by the National Ethics Committee for Health Research of the Ministry of Health of Cambodia (117/NECHR).

#### 3.1.1. Anthropometric Measurements

Weight, length/height and mid-upper arm circumference were measured in duplicate for each child to reduce measurement bias. Weights were recorded using calibrated digital balances (SECA, Germany) with 100 g precision. Recumbent lengths or standing heights were measured to the nearest 1 mm using UNICEF boards with standing plates and moveable head boards. Anthropometric variables were calculated according to the WHO Growth Child standards 2006 [27,28] as HAZ and WHZ, which values were plotted to examine the presence of implausible values and recorded as missing when WHZ were below −5 or above +5 or HAZ were below −6 or above +6 [27]. The UNICEF standard MUAC tape (S0145620 MUAC, Child 11.5 Red/PAC-50) was used to record the child’s mid-upper arm circumference with cut-off points from red to yellow at 115 mm and from yellow to green at 125 mm. The nutritional statues of the child represent the main outcomes of the analyses measured as continuous variables: WHZ or MUAC to define wasting, and HAZ for stunting.

#### 3.1.2. Water, Sanitation and Hygiene Measurements

The child-sensitive WASH composite score, including water, sanitation and hygiene components, represented the independent variable of the analyses. Water, sanitation and hygiene practices of the study population were simultaneously collected along the three years study leading to consider WASH practices as one variable, equally weighting the three components.

Two WASH composites scores were independently created: the JMP-Child-sensitive (JMP-CS) composite score referring to the worldwide classification used by the WHO/UNICEF Joint Monitoring Programme [29] and readapted as child-sensitive score by the authors; and the National-Child-sensitive (National-CS) composite score defined according to the Cambodia Demographic Health survey [2], the socio-economic survey [30] and the brief developed jointly by Water and Sanitation Programme and UNICEF as part of a series of country profiles about sanitation for children [31]. Households and child WASH practices were used to create JMP-CS or National-CS water, sanitation and hygiene sub-scores which were equally weighted to subsequently create the respective composite scores. Each sub-score was independently created by selecting variables and classifying the values according to either guideline.

JMP-CS classification—Two variables were selected to define the water sub-score: main source of water (assumed to be functional at time of survey) and location of water source. Two variables for the sanitation sub-score: type of toilet and strategy used to dispose child excreta. Although the child’s disposal of excreta is not a component of the JMP classification, the variable was used to create a child-sensitive score according to the service level of classification of sanitation for adults [32]. Furthermore one recent study in Ethiopia highlighted the importance to ensure safe and improved child stool disposals [33]. One variable was used for the hygiene sub-score: whether the caretaker washes their hands after cleaning the child’s bottom and/or after moving faeces. Handwashing linked with child defecation and excreta management is not a component of the JMP classification and national surveys. However, washing hands after using the toilet is a worldwide and nationally accepted advanced level of hygiene that has been used to create a child-sensitive hygiene sub-score. National-CS classification—Five variables were used to define the water sub-score: main source of water (assumed to be functional at time of survey), location of water source, frequency of water treatment, method used to treat the water and frequency of child’s water treatment. Two variables for the sanitation sub-score: type of toilet and strategy used to dispose child excreta. Similarly to the JMP-CS, one variable was used to create the hygiene sub-score: whether the caretaker washes their hands after cleaning the child’s bottom and/or after moving faeces.

For each variable the values were classified as “improved” (coded 1) or “not improved” (coded 0) according to either guideline as shown in Figure 1. By weighting equally the chosen variables, JMP-CS or National-CS sub-scores were created, each one granting a maximum score of 1 for the most improved and a minimum score of 0 for the non-improved water, sanitation or hygiene services.

Finally a total of six child-sensitive WASH composite scores, defined as continuous variables ranging from 0 to 1, were created by summing water, sanitation and hygiene sub-scores weighed equally: JMP-CS or National-CS composite scores at baseline, follow-up 2 and at follow-up 3.

#### 3.1.3. Variable Included in the Analyses

The child’s age in months was calculated by subtracting the date of the visit from the date of birth of the child; it was considered as a continuous variable for the analyses. Gender, exclusive breastfeeding and mother education were considered as binary variables and province as categorical variable uses Phnom Penh as the reference province compared to Kratie and Ratanakiri. A child dietary diversity score was created for baseline, follow-up 2 and 3 according to the Food and Agriculture Organization guidelines for measuring household and individual dietary diversity [34]. The score was based on recall by mothers/caregivers of children’s consumption of nine food groups within the past 24 h. The variable was treated as continuous for the analyses. The household wealth index is a composite measure of a household’s living standard that was calculated through principal component analysis [35]. It gathers information regarding employment status and ownership of a car, television, generator, radio, computer bicycle, motorcycle, boat with and without motor, phone, and refrigerator. The wealth index was then divided into quintiles and treated as continuous.

#### 3.1.4. Statistical Analyses

The analyses were conducted using STATA/IC Release 15.1 (Stata Corp., College Station, Texas, USA). Significance was defined as *p* < 0.05. The database was created by merging (one to one on “ChildID” key variable) baseline, follow-up 2 and follow-up 3 datasets.

Missing data were enumerated for all variables of interest before analyses. Observations were unbalanced because they were not taken at regular times. Missing values were considered as missing at random, automatically corrected for in mixed-effects models.

Anthropometric measures and prevalences were calculated at each study visit. Children from 0 to 59.9 months were classified as wasted when the WHZ was inferior to −2 standard deviations (SD) of the WHO Child Growth Standards median; whereas children from 6 to 59.9 months were classified as wasted when the WHZ was inferior to −2 standard deviations (SD) of the WHO Child Growth Standards median and/or the MUAC was inferior to 125 mm. Children from 0 to 59.9 months were classified as stunted when the HAZ was inferior to −2 SD. Two-sample Z-tests were used to compare proportions of wasting and stunting prevalence between consecutive study visits.

Due to the repeated measures along the study of most of the variables considered, mixed-effects models were used to perform univariate and multivariate analyses. The dataset was re-shaped as long where the subject’s repeated responses is in a single row and each response is in a separate column. Mixed-effects linear regression models were performed using the maximum likelihood method, whereby the unique child identifier represents the random part of the model. Both fixed and random parts of the model include the time variable. Crude analyses were performed to investigate the association between each outcome of interest (WHZ and MUAC for wasting and HAZ for stunting) and the JMP-CS or National-CS composite scores (independent variables). Similarly, the relationship between each outcome and each possible confounder was also investigated; the same was done for the relationship between each exposure and each possible confounder. The variables statistically significant associated with exposure and outcomes were included as confounders into the multivariate analyses. Mixed-effects linear regression models were used firstly to analyse the relationship between outcomes and exposures, adjusted for one confounder at the time. The full models were then fit using all confounders together: gender and age by default, plus exclusive breastfeeding, dietary diversity score, mother education, wealth index and province, selected on significance.

Linear mixed-effects models entail two assumptions: (i) homoscedasticity and (ii) that the errors are normally distributed. These assumptions were checked by (i) plotting the residuals against the linear prediction from fixed plus random effects values and (ii) creating histograms of the fixed, fixed plus random effects and residual values. Finally, multilevel mixed-effects generalized linear models were used as alternative approach.

## 4. Results

### 4.1. General Characteristics of Participants

A total of 5328 children from 5014 households were enrolled in the longitudinal study. 33% (1768/5328) came from Phnom Penh, 36% (1939/5328) from Kratie Province and 31% (1621/5328) from Ratanakiri Province. The mean child’s age was 11.8 ± 7 months at baseline, 16.2 ± 8 months at follow-up 2 and 19.8 ± 9 at follow-up 3, with similar gender distribution of 2656 males and 2654 females (18 missing values). The distribution of household wealth index was slightly skewed with 20% (1067/5328) of the population belonging to the poorest quintile, 21% (1110/5328) to the second poorest 23% (1223/5328) to the middle, 15% (797/5328) to the second richest and 19% (1031/5328) to the richest (100 missing values) and 61% (3260/5328) of the mothers had no–low education (227 missing values).

### 4.2. Anthropometric Characteristics of Participants

Wasting prevalence for children from 0 to 59.9 months (WHZ < −2 SD) decreased from 14.4% (624/4343) at baseline to 8.5% (293/3441) at follow-up 2 (*p* < 0.01) and did not significantly change at follow-up 3 with 9.1% (305/3357) (*p* = 0.4), whereas for children from 6.0 to 59.9 months (WHZ < −2 SD and/or MUAC < 125 mm) wasting prevalence decreased from 20% (649/3241) at baseline to 12.3% (368/2987) at follow-up 2 (*p* < 0.01) without significantly decreasing at follow-up 3 with 11.7% (368/3151) (*p* = 0.5). The prevalence of stunting increased along the study from 19.2% (836/4351) at baseline to 25.9% (893/3445) at follow-up 2 (*p* < 0.01) to 28.6% (961/3358) at follow-up 3 (*p* = 0.01). Anthropometric characteristics are reported in Table 1.

### 4.3. Water, Sanitation and Hygiene Characteristics of Participants

JMP-CS and National-CS composite scores showed respectively a mean of 0.68 ± 0.19 and 0.68 ± 0.16 at baseline, 0.74 ± 0.21 and 0.76 ± 0.18 at follow-up 2 and 0.72 ± 0.19 and 0.76 ± 0.18 at follow-up 3. Considering sub-scores individually, the JMP-CS and National-CS sanitation sub-scores showed the lowest means, respectively 0.53 ± 0.33 and 0.37 ± 0.36 at baseline, 0.6 ± 0.35 and 0.45 ± 0.4 at follow-up 2 and 0.63 ± 0.34 and 0.45 ± 0.4 at follow-up 3. JMP-CS and National-CS water sub-scores means were respectively 0.53 ± 0.37 and 0.74 ± 0.2 at baseline, 0.63 ± 0.39 and 0.78 ± 0.0.23 at follow-up 2 and 0.53 ± 0.36 and 0.77 ± 0.23 at follow-up 3. Finally the JMP-CS and National-CS hygiene sub-score were always very high with 0.99 ± 0.09 and 0.99 ± 0.09 at baseline, 0.998 ± 0.4 and 0.998 ± 0.4 at follow-up 2 and 0.996 ± 0.0.06 and 0.996 ± 0.0.06 at follow-up 3.

### 4.4. Univariate and Multivariate Analyses for Wasting

Table 2 shows the results of univariate and multivariate analyses for WHZ and MUAC. A strong crude positive association (*p* < 0.001) was found with each independent variable (JMP-CS or National-CS composite score). For each unit increased in the JMP-CS composite score, WHZ increases by 0.28 SDs (95%CI 0.18–0.38, *p* < 0.001) and MUAC by 0.54 cm (95%CI 0.43–0.65, *p* < 0.001); whereas for each unit increased in the National-CS composite score, the results show an increase of 0.55 SDs (95%CI 0.4–0.69, *p* < 0.001) for WHZ and 0.95 cm (95%CI 0.78–1.12, *p* < 0.001) for MUAC. All the investigated variables (gender, age, exclusive breastfeeding, dietary diversity intake, mother education, wealth index, province) show evidence for an association with WHZ or MUAC (*p* < 0.05). A positive relationship with WHZ was observed for gender, mother education and wealth index and a negative relationship was found with age, consumption of food, dietary diversity score and province of residence. Gender and province of residence were negatively related with MUAC, whether all the other variables showed positive relationship. Significant associations were found for all variables except for gender showing no evidence for an association with the JMP-CS (95%CI −0.02–0.001, *p* = 0.065) and the National-CS (95%CI −0.02–0.01 *p* = 0.473) composite scores. Evidence of association (*p* < 0.05) between WHZ or MUAC and JMP-CS or National-CS composite score was found adjusting for one confounder at the time (data not shown).

The full models were then fit using all confounders together: gender and age by default, exclusive breastfeeding, dietary diversity score, mother education, wealth index and province, selected on significance. Table 2 shows the adjusted analyses using JMP-CS and National-CS composite score as independent variable and WHZ or MUAC as dependent variables.

After adjusting for all confounders, there was a remained significant association between the JMP-CS composite scores and WHZ or MUAC (*p* < 0.05). Compared to the crude analyses, the WHZ-JMP-CS composite score coefficient decreased to 0.14 SDs per unit JMP-CS composite score increase and the MUAC-JMP-CS composite score coefficient to 0.18 cm. The fully adjusted models using the National-CS composite score also showed a significant association with both WHZ and MUAC: after adjusting for all confounders, either the coefficient related to WHZ decreased to 0.4 SDs and MUAC to 0.4 cm.

### 4.5. Univariate and Multivariate Analyses for Stunting

Table 3 shows the results of univariate and multivariate analyses for HAZ. A strong crude positive association (*p* < 0.01) was found with each independent variable (JMP-CS or National-CS composite score). For each unit increased in the JMP-CS composite score HAZ increases by 0.21 SDs (95%CI 0.13–0.29, *p* < 0.01) whereas an increase of 0.35 SDs (95%CI 0.22–0.47, *p* < 0.01) for HAZ was observed for each unit increase in the National-CS composite score. Gender, age, exclusive breastfeeding, dietary diversity intake, mother education, wealth index and province show evidence for an association with HAZ (*p* < 0.05). Similar to WHZ, a positive relationship was observed for gender, mother education and wealth index and a negative relationship was found with age, consumption of food, dietary diversity score and province of residence. The full models were then fit using all confounders together: gender and age by default, plus exclusive breastfeeding, dietary diversity score, mother education, wealth index and province, selected on significance. Table 3 shows the adjusted analyses using JMP-CS and National-CS composite score as independent variable and HAZ as dependent variables. After adjusting for all confounders, the association between the JMP-CS composite scores and HAZ remain significant (*p* < 0.05). Compared to the crude analyses, the HAZ-JMP-CS composite score coefficient decreased to 0.15 SDs per unit JMP-CS composite score increase. This association was weaker when the National-CS composite score was used (*p* = 0.081).

## 5. Model Diagnostics

The histograms of fixed plus random effects and residual values indicated that they are normally distributed. Figure 1, shows for each model the plot of the residuals against the linear prediction from fixed plus random effects values showing mild heteroscedasticity. Given the presence of mild heteroscedasticity, multilevel mixed-effects generalized linear models were used as an alternative approach for the six models (data not shown). Similar estimates were produced for the univariate analyses showing a strong crude positive association (*p* < 0.01) between each independent variable (JMP-CS or National-CS composite score) and each dependent variable (WHZ, MUAC or HAZ). Similar results were obtained for the fully adjusted models, except for the association between the JMP-CS composite score and WHZ, which became weaker (*p* = 0.069).

## 6. Discussion

To our knowledge, this study is the first to investigate the longitudinal association between water, sanitation and hygiene conditions, regrouped as child-sensitive WASH composite scores, and the nutritional status of children in Cambodia. Data were collected in a longitudinal cohort study in six districts localized in Phnom Penh and Kratie and Ratanakiri, two north-eastern provinces of Cambodia. The study, started in 2016 and ending in March 2019, aims to collect in-depth data to monitor the progress on health, nutrition, WASH practices, and cognitive development of children under five years of age and to inform the government about the progress that can be made with enhanced health monitoring. Although UNICEF did not carry out interventions in the targeted districts, we cannot exclude that other development partners did. The analyses performed in this paper take into account participants’ changes over time: due to external or unknown causes, participants might have changed their main source of water or type of toilet used over time or their hygiene habits. Decreasing of wasting from baseline to follow-up 2 and increasing of stunting prevalence along the study were observed. The increase of stunting prevalence can be explained by the chronicity of the nutrition condition. Stunting is a largely irreversible condition: a child cannot recover height in the same way that they can regain weight. Multiple factors as poor maternal health and nutrition, inadequate infant and young child feeding practices and infection cause stunting [36] which are more difficult to tackle in its entirety. Without targeted interventions, stunted children are more likely to remain stunted and the new cases explain the increased prevalence observed along the study. Actions in multiple areas are required to achieve the Global Nutrition Target 2025 of 40% reduction in the number of children under five years of age who are stunted.

The crude and adjusted—for gender, age, exclusive breastfeeding, dietary diversity score, mother education, wealth index and province of residence—analyses showed evidence for a positive association between child-sensitive WASH composite scores and the child anthropometry indicators for wasting. Regardless of the nutritional status of the child, by improving one unit of the JMP-CS composite score, the child’s WHZ increases by 0.14 SDs and the child’s MUAC by 0.18 cm. Similarly, improving one unit of the National-CS composite score, the child’s WHZ increases by 0.4 SDs and the child’s MUAC by 0.4 cm. However, the fully adjusted association between age or gender and WHZ took opposite directions compared to the fully adjusted association with MUAC. A recent study in the same cohort of Cambodian children showed that both MUAC and WHZ showed gender bias, with MUAC identifying more girls and WHZ identifying more boys with acute malnutrition. The gender bias, strongest for MUAC, diminished with older age, but remained significant up to 30 months of age [37]. The results obtained in the cohort stressed on the importance to maintain both WHZ and MUAC for the identification of acute malnutrition in children in Cambodia.

Moreover crude and adjusted analyses showed evidence for a positive association between the JMP-CS composite score and the child anthropometry indicator for stunting. Regardless the nutritional status of the child, by improving one unit of the JMP-CS composite score, the child’s HAZ increases by 0.15 SDs. However, this association is weaker if the National-CS score is used (*p* = 0.08), albeit the crude analysis was strongly associated.

This study adds to the limited evidence on the direct association between WASH practices including water, sanitation and hygiene, and both wasting and stunting undernutrition statues. To date, mainly randomized controlled trials explored this association, showing discordant results. Three RCTs in Indonesia, India and Mali showed stunting reduction and child height and weight improvements after sanitation interventions [15,16,17]; and similarly the systematic review and meta-analysis of Dangour et al., showed slight but significant benefits of water quality and hygiene interventions on the child’s length growth [13]. However, two recent RCTs in Bangladesh and Kenya showed that integration of water, sanitation, and handwashing with nutrition did not have any benefit on children’s growth [19,20].

This study brings evidence for a positive longitudinal association in the six targeted districts of Cambodia, whereby various factors of causality were accounted for, minimizing the potential residual confounding and collinearity. The large sample size (*n* = 5238) increases the power of the statistical analyses, but may increase study attrition and missing data. Indeed, several mothers and children did not show up at each study visit. This could be due to migration often occurring in the country, where families move across provinces seeking for seasonal jobs. The method of analysis chosen adjusts for missing data, as assuming they are missing at random. Furthermore, tablet-based data collection automatically entering information into a database, allows for data checking along the course of the collection and for recover data when they are missing. Quality data collection was ensured during the study. Anthropometry data (length, weigh and MUAC) were measured in duplicate reducing measurement bias. These data were also recorded on paper in order to calculate the z-scores in situ and detect and refer severely malnourished children for treatment. Paper-based data collection also allows having a backup in case of any mistake occurred in the tablet data entry. The WASH questions were formulated in an understandable manner supported by figures showing the different type of source of water (piped, dug well, pumping water, river…) and toilet (different type of flush/pour flush toilet, pit latrine with/without slab, bucket, field…), reducing information bias.

However, although only comparable variables among the study visits were used for the analysis, the questionnaire slightly changed after baseline, potentially biasing measurements. As shown in Appendix A, the answers of the WASH practices questions were phrased slight differently in baseline but the values were referable to “improved” or “not improved” classification according to the guidelines. Furthermore we acknowledge weakness on the available variables used to create water, sanitation and hygiene sub-scores. Complete water quality and quantity data including indicators for accessibility (within 30 min, on premises), availability when needed and water quality, such as microbiological and chemical contaminations, were collected for only around 800 randomly selected households. These data could not be used in the longitudinal analyses because of the small proportion compared to the whole cohort and will require further analysis. The sanitation sub-scores did not include the variable indicating whether facilities were shared with other households because collected using inconsistent classifications across time. The hygiene sub-scores lack to gather information on adults’ hygiene practices after toileting which may have impacts on food preparation and other activities that impact the child. Only child-related hygiene practices were collected in the study, leading to limit the analyses on available data. Finally the indicator about handwashing with water and/or soap was not comparable between baseline and follow-ups and it was thus excluded for the creation of the hygiene sub-scores.

The longitudinal study aims to inform the government about the progress in the targeted districts underlining the importance to consider the Cambodian context in the analyses. Indeed, two CS-composite scores were independently created: the JMP-CS defined according to the worldwide accepted JMP guidelines and readapted as child-sensitive score by the authors; and the National-CS composite score, according to the national guidelines more conservative for water quality and sanitation classifications. The national drinking water classification adds an indicator about the method used for water treatment and considers bottled water as “not improved” because of the high numbers of contaminated bottles sold in the country, probably due to unsafe management (Poirot et al., submitted to Maternal and Child Nutrition). The national sanitation ladder considers buried faeces as “not improved” because of the frequent floods occurring in the country, especially in the rainy season, meaning buried stools easily re-surface. We obtained similar results for wasting highlighting the importance to integrate nutrition and WASH implementing programmes to improve the growth of children in the targeted districts. The association with stunting is significantly associated with the CS-JMP composite score but it becomes less significant when the National-CS composite score was used as independent variable. Compared to the JMP-CS composite score, the National-CS composite score included more variables leading to increased missing data. Consequently, the final models were performed on a smaller sample size decreasing the statistical power of the analyses. This may explain the differences obtained for stunting and will enforce the results obtained for wasting. Our findings are not generalizable for the whole country but rather to the selected districts where all villages were targeted.

## 7. Conclusions

The findings of this study show a strong significant association between water, sanitation and hygiene conditions and wasting and stunting in Cambodian children residing in the targeted districts. These results aim to support the government and other actors to design and mobilise resources for integrated WASH and nutritional interventions, especially in the north-eastern provinces where the prevalence of stunting and wasting is higher than the urban provinces [2] and half of the population still use open defecation (WHO/UNICEF, 2017c).

## Figures and Tables

**Figure 1 nutrients-11-02142-f001:**
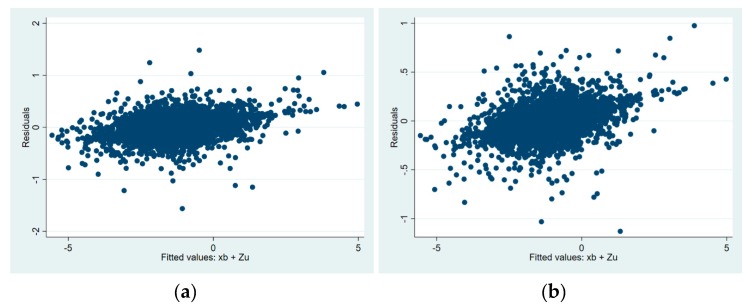
Model diagnosis for the six full adjusted models. Plot of the residuals against the linear prediction from fixed plus random effects values for WHZ_JMP-CS (**a**) and National-CS (**b**) composite score, MUAC_JMP-CS (**c**) and National-CS (**d**) composite score and HAZ_JMP-CS (**e**) and National-CS (**f**) composite score.

**Table 1 nutrients-11-02142-t001:** Description of study population: age, gender, anthropometric measurements by study visit.

	Baseline	Follow-Up 2	Follow-Up 3
**Female**	2654/5310 (50%)	2654/5310 (50%)	2654/5310 (50%)
**Age of study population (months) ^1^**	11.8 ± 7	16.2 ± 8.5	19.8 ± 8.9
**WHZ of study population ^1^**	−0.8 ± 1.1	−0.7 ± 1	−0.8 ± 1
*Normal*	3719/4343 (85.6%)	3148/3441 (91.5%)	3052/3357 (90.9%)
*Wasted (WHZ < –2 SD)*	624/4343 (14.4%)	293/3441 (8.5%)	305/3357 (9.1%)
**MUAC of study population (6.0–59.9 months) ^1^**	13.8 ± 1.2	14 ± 1.1	14.1 ± 1.1
*Normal*	2914/3239 (90%)	2786/2981 (93.5%)	2969/3149 (94.3%)
*Wasted (MUAC < 125 mm)*	325/3239 (10%)	195/2981 (6.5%)	180/3149 (5.7%)
**HAZ of study population ^1^**	−1 ± 1.3	−1.3 ± 1.1	−1.4 ± 1.1
*Normal*	3515/4351 (80.8%)	2552/3445 (74.1%)	2397/3358 (71.4%)
*Stunted (HAZ < –2SD)*	836/4351 (19.2%)	893/3445 (25.9%)	961/3358 (28.6%)

^1^ Mean ± Standard Deviation.

**Table 2 nutrients-11-02142-t002:** JMP-CS and National-CS composite scores, gender, age, exclusive breastfeeding, dietary diversity intake, mother education, wealth index and province in relation to the weight-for-height z-scores and mid-upper arm circumference over time.

**Weight-for-Height z-Score**									
	**Crude Analyses**			**Adjusted Analyses (JMP-CS)**			**Adjusted Analyses (National-CS)**		
	**Estimate**	**95% CI**	***p*-Value**	**Estimate**	**95% CI**	***p*-Value**	**Estimate**	**95% CI**	***p*-Value**
**JMP-CS composite score**	0.28	0.18–0.38	<0.001	0.14	0.01–0.27	0.03			
**National-CS composite score**	0.55	0.4–0.69	<0.001				0.4	0.2–0.58	<0.001
**Gender**									
*Male*	ref.								
*Female*	0.06	0.002–0.11	0.04	0.06	0.004–0.12	0.051	0.08	0.01–0.15	0.021
**Age**	−0.016	−0.02–−0.012	<0.001	−0.028	−0.03–−0.02	<0.001	−0.028	−0.03–−0.02	<0.001
**Exclusive breastfeeding**									
*Exclusive breastfeeding*	ref.								
*Other food/drinks*	−0.27	−0.33–0.21	<0.001	−0.35	−0.43–0.27	<0.001	−0.35	−0.45–0.25	<0.001
**Dietary diversity intake**	−0.02	−0.03–−0.007	0.001	0.01	−0.006–0.02	0.268	0.01	−0.01–0.02	0.516
**Mother education**									
*No-low education*	ref.								
*Higher education*	0.22	0.16–0.28	<0.001	0.1	0.04–0.18	0.001	0.1	0.001–0.15	0.052
**Wealth index**	0.14	0.12–0.16	<0.001	0.09	0.07–0.12	<0.001	0.1	0.07–0.12	<0.001
**Province**									
*Phnom Penh*	ref.								
*Kratie*	−0.4	−0.5–0.34	<0.001	−0.31	−0.4–0.22	<0.001	−0.24	−0.34–0.14	<0.001
*Ratanakiri*	−0.35	−0.4–0.3	<0.001	−0.16	−0.25–0.06	0.001	−0.07	−0.17–0.04	0.234
**Mid-Upper Arm Circumference**									
	**Crude analyses**			**Adjusted analyses (JMP-CS)**			**Adjusted analyses (National-CS)**		
	**Estimate**	**95% CI**	***p*-value**	**Estimate**	**95% CI**	***p*-value**	**Estimate**	**95% CI**	***p*-value**
**JMP-CS composite score**	0.54	0.43–0.65	<0.001	0.18	0.04–0.31	0.013			
**National-CS composite score**	0.95	0.78–1.12	<0.001				0.4	0.16–0.58	0.001
**Gender**									
*Male*	ref.								
*Female*	−0.36	−0.42–0.29	<0.001	−0.39	−0.45–−0.32	<0.001	−0.39	−0.46–0.31	<0.001
**Age**	0.04	0.037–0.05	<0.001	0.02	0.017–0.03	<0.001	0.02	0.016–0.3	<0.001
**Exclusive breastfeeding**									
*Exclusive breastfeeding*	ref.								
*Other food/drinks*	0.61	0.54–0.68	<0.001	0.55	0.46–0.63	<0.001	0.51	0.4–0.6	<0.001
**Dietary diversity intake**	0.07	0.05–0.08	<0.001	0.02	0.002–0.03	0.027	0.01	−0.003–0.03	0.102
**Mother education**									
*No-low education*	ref.								
*Higher education*	0.33	0.27–0.4	<0.001	0.14	0.06–0.21	<0.001	0.1	0.03–0.2	0.008
**Wealth index**	0.19	0.17–0.22	<0.001	0.1	0.08–0.14	<0.001	0.1	0.08–0.14	<0.001
**Province**									
*Phnom Penh*	ref.								
*Kratie*	−0.52	−0.6–0.45	<0.001	−0.28	−0.37–0.18	<0.001	−0.24	−0.35–0.13	<0.001
*Ratanakiri*	−0.65	−0.73–0.58	<0.001	−0.38	−0.48–0.28	<0.001	−0.31	−0.43–0.2	<0.001

**Table 3 nutrients-11-02142-t003:** JMP-CS and National-CS composite scores, gender, age, exclusive breastfeeding, dietary diversity intake, mother education, wealth index and province in relation to the height-for-age z-scores over time.

**Height-for-Age Z-Score**									
	**Crude Analyses**			**Adjusted Analyses (JMP-CS)**			**Adjusted Analyses (National-CS)**		
	**Estimate**	**95% CI**	***p*-Value**	**Estimate**	**95% CI**	***p*-Value**	**Estimate**	**95% CI**	***p*-Value**
**JMP-CS composite score**	0.21	0.13–0.29	<0.001	0.15	0.04–0.25	0.006			
**National-CS composite score**	0.35	0.22–0.47	<0.001				0.15	−0.02–0.32	0.081
**Gender**									
*Male*	ref.								
*Female*	0.1	0.04–0.17	0.002	0.12	0.5–0.18	0.001	0.11	0.03–0.19	0.004
**Age**	−0.025	−0.03–-0.02	<0.001	−0.03	−0.04–−0.029	<0.001	−0.038	−0.04–−0.03	<0.001
**Exclusive breastfeeding**									
*Exclusive breastfeeding*	ref.								
*Other food/drinks*	−0.18	−0.23–−0.12	<0.001	−0.07	−0.13–0.003	0.04	−0.11	−0.2–-0.03	0.009
**Dietary diversity intake**	−0.02	−0.03–−0.008	<0.001	−0.01	−0.02–−0.003	0.012	−0.01	−0.02–0.004	0.173
**Mother education**									
*No-low education*	ref.								
*Higher education*	0.4	0.33–0.47	<0.001	0.18	0.1–0.26	<0.001	0.17	0.09–0.26	<0.001
**Wealth index**	0.19	0.16–0.2	<0.001	0.13	0.1–0.16	<0.001	0.13	0.09–0.16	<0.001
**Province**									
*Phnom Penh*	ref.								
*Kratie*	−0.41	−0.49–−0.34	<0.001	−0.14	−0.24–−0.04	0.007	−0.16	−0.26–-0.05	0.005
*Ratanakiri*	−0.62	−0.7–−0.54	<0.001	−0.36	−0.46–−0.26	<0.001	−0.37	−0.49–-0.25	<0.001

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
