# Peer review of "Child-Sensitive WASH Composite Score and the Nutritional Status in Cambodian Children"

_nutrients, 2019, doi:10.3390/nu11092142_

Round 1

Reviewer 1 Report

This paper presents findings from a correlational analysis of HAZ and WHZ (and MUAC) as predicted by composite WASH behavioral index scores estimated by the authors.

The paper is pretty straight-forward so i only have a couple of comments for the authors:

I think it would be helpful to explain why it's necessary to create this composite score rather than just looking at multivariate relationships between the dependent variables (nutrition status) and the WASH characteristics. All of the results are presented with respect to different magnitudes of changes in the WASH composite scores, but it's not clear to me what that translates to in practical terms (e.g. how much does the score move if my family upgrades our sanitation facility?).

Author Response

Point 1: I think it would be helpful to explain why it's necessary to create this composite score rather than just looking at multivariate relationships between the dependent variables (nutrition status) and the WASH characteristics.

Response 1: It would have been possible to look at multivariate relationships between the nutrition status and the WASH characteristics. However, in this study we aim to define the WASH characteristics as one variable, equally weighting its components as water, sanitation and hygiene. Unlikely to most of the studies present in the literature, the longitudinal study simultaneously collected information about water, sanitation and hygiene practices of the study population along three years. Therefore we were able to define the composite score and sub-scores of the study population along the study.

Point 2: All of the results are presented with respect to different magnitudes of changes in the WASH composite scores, but it's not clear to me what that translates to in practical terms (e.g. how much does the score move if my family upgrades our sanitation facility?). 

Response 2: One unit increase of child sensitive WASH composite is positively associated with the child anthropometric indicators for wasting and stunting. In practical terms by increasing the WASH practices the child anthropometric indicators are more likely to be improved. This analysis aims to demonstrate a positive association between WASH composite scores and the child anthropometric indicators. We acknowledge that further studies are necessary to define which of the three components would lead to a better improvement of the child nutritional status.

Reviewer 2 Report

The manuscript is presented very well. The research and analysis methods are sound and the results are very valuable and well discussed. I wish to invite the authors to consider a recently published work which is closed to their current work. The article is accessible here: https://doi.org/10.1186/s12889-019-7325-9

What kind of weak points does this newly published paper have which your work could cover it? Maybe it worth considering it in your literature review?

Author Response

Point 1: I wish to invite the authors to consider a recently published work which is closed to their current work. The article is accessible here: https://doi.org/10.1186/s12889-019-7325-9

What kind of weak points does this newly published paper have which your work could cover it? Maybe it worth considering it in your literature review?

Response 1: Thank you for suggesting this new published article that was not available when the manuscript was submitted. The rreference has been added into the "Water, sanitation and hygiene measurements paragraph" to highlith the importance to consider the “stool disposal” variable to create the sub-score. However as Sahiledengle et al. do not take into account the nutritional status of children, it may be difficult to highlight weak points when compared to our study.
